# Comparison of samplers collecting airborne influenza viruses: 1. Primarily impingers and cyclones

**Peter C. Raynor** [ID]**[1]\***, **Adepeju Adesina[1], Hamada A. Aboubakr** [ID]**[2], My Yang[2], Montserrat Torremorell[2], Sagar M. Goyal** [ID]**[2]**

**1** Division of Environmental Health Sciences, University of Minnesota, School of Public Health, Minneapolis, Minnesota, United States of America, **2** University of Minnesota, College of Veterinary Medicine, Veterinary Population Medicine Department, St. Paul, Minnesota, United States of America

\* praynor@umn.edu

**Data Availability Statement:** All relevant data are within the manuscript and its Supporting Information files.

## Abstract

Researchers must be able to measure concentrations, sizes, and infectivity of virus-containing particles in animal agriculture facilities to know how far infectious virus-containing particles may travel through air, where they may deposit in the human or animal respiratory tract, and the most effective ways to limit exposures to them. The objective of this study was to evaluate a variety of impinger and cyclone aerosol or bioaerosol samplers to determine approaches most suitable for detecting and measuring concentrations of virus-containing particles in air. Six impinger/cyclone air samplers, a filter-based sampler, and a cascade impactor were used in separate tests to collect artificially generated aerosols of MS2 bacteriophage and swine and avian influenza viruses. Quantification of infectious MS2 coliphage was carried out using a double agar layer procedure. The influenza viruses were titrated in cell cultures to determine quantities of infectious virus. Viral RNA was extracted and used for quantitative real time RT-PCR, to provide total virus concentrations for all three viruses. The amounts of virus recovered and the measured airborne virus concentrations were calculated and compared among the samplers. Not surprisingly, high flow rate samplers generally collected greater quantities of virus than low flow samplers. However, low flow rate samplers generally measured higher, and likely more accurate, airborne concentrations of Infectious virus and viral RNA than high flow samplers. To assess airborne viruses in the field, a two-sampler approach may work well. A suitable high flow sampler may provide low limits of detection to determine if any virus is present in the air. If virus is detected, a suitable lower flow sampler may measure airborne virus concentrations accurately.

## Introduction

Substantial evidence exists for key instances of airborne virus transmission. Sattar et al. [1] reported several cases of aerosol transmission for many types of human and animal viruses. In a review article, Musher [2] found that influenza virus, adenovirus, respiratory syncytial virus (RSV), and rhinovirus could all be transmitted among humans via large or medium-sized aerosol

**Funding:** PCR: This research was conducted through the Upper Midwest Agricultural Safety and Health (UMASH) Center, which is funded by cooperative agreement U54 OH010170 from the National Institute of Occupational Safety and Health, http://www.cdc.gov/niosh. The funders had no role in study design, data collection and analysis, decision to publish, or preparation of the manuscript.

**Competing interests:** The authors have declared that no competing interests exist.

droplets. However, only influenza virus and adenovirus could be transmitted by smaller droplets capable of staying airborne for long periods. Laboratory studies indicate that influenza viruses can also be transmitted to animals through air [3–5]. Many emerging influenza viruses are thought to be able to transmit through air among people or animals and between animals and people, or to have the potential to develop this transmissibility [6–9]. Most recently, the SARS-CoV-2 coronavirus has been found to be transmissible through the air [10].

Emerging zoonotic viruses such as H5N1 avian influenza virus, novel 2009 H1N1 influenza virus, 2012 H3N2 variant influenza virus, H7N9 avian influenza virus, and H5N2 and H5N8 avian influenza viruses have posed real or potential risks to veterinary, swine, and poultry workers [11–18]. Swine, in particular, may pose particular risks because they can be infected by avian, swine, and human viruses, potentially leading to genetic reassortment that can produce viruses of potential pandemic [19]. The Occupational Health and Safety Administration (OSHA) considers poultry workers, veterinarians, animal handlers, pet shop workers, and zookeepers to be at risk of exposure to avian influenza viruses [20]. OSHA has also provided guidance to pork producers about protecting swine workers in their facilities from swine influenza viruses [21].

Animals in agricultural operations generate particles that are small enough to be transported substantial distances, and many of these particles contain viral RNA [22–24]. However, little information exists regarding the concentrations and sizes of airborne particles that contain infectious viruses in animal agriculture [22]. In addition, influenza viruses in aerosols can remain infectious for long periods of time [25]. These studies indicate that it is important to evaluate the infectivity of airborne viruses. To know how far virus-containing particles may travel through air, assess where they may deposit in the respiratory system if inhaled by workers, determine if they can cause illness, and identify the most effective methods for limiting exposure to the particles, we must be able to measure concentrations, sizes, and infectivity of virus-containing particles present in animal production facilities.

Viral and other biological aerosols may be sampled from air using a variety of different approaches [26–30]. Impingers pass sampled air through one or more air nozzles into a liquid collection medium in which particles may be captured as air bubbles through the liquid. Cyclones use the inertia of particles in air rotating through a passage to capture the particles on the surface of the passage via centrifugal forces. Some samplers combine impingement and cyclonic collection in a single sampler. Impactors utilize particle inertia to collect particles by turning a particle-filled airstream so that larger particles with higher inertia are captured on a surface. An instrument that uses a series of impactors to progressively capture different particle sizes, from large to small particles, is referred to as a cascade impactor. Filters are a primary method of sampling non-biological particles but can be used to sample biological aerosols as well. Electrostatic samplers apply charge to particles by generating ions in a sampled airstream and then collect the charged particles in an electrical field. The relative effectiveness of these different sampling technologies at measuring concentrations of viral RNA and infectious virus has not been rigorously characterized.

A few studies have evaluated the ability of samplers to collect airborne influenza viruses, either experimentally or in the field. Fabian et al [31]. compared an impinger that incorporated cyclonic airflow (the SKC Biosampler), Teflon filters, gelatin filters, and a cascade impactor at collecting an experimentally generated aerosol of H1N1 influenza virus. These authors found that the impinger/cyclone measured the highest concentrations of viral RNA, followed by the filters, and then the impactor, and that the impinger/cyclone measured much higher concentrations of infectious virus than the other samplers. Ge et al [32]. used an eight-stage non-viable Andersen cascade impactor to evaluate the survivability of laboratory-generated airborne viruses over long sampling periods. When comparing sampling periods of 1 and 6 hours, collected viruses, including influenza viruses, were significantly inactivated in the impactor,

suggesting that a longer sampling duration may not lead to better detection of infectious viruses in field measurements. A wet cyclonic collector has been used successfully to measure airborne influenza RNA in swine housing, although success at detecting infectious virus has been mixed [33,34]. Chen et al. [35] found that sampling with Teflon filters in closed-face cassettes could be used to effectively measure influenza virus RNA in wet poultry markets.

Several studies have measured the sizes of airborne particles with which viral RNA is associated. Using a two-stage cyclone sampler, Blachere et al. [36] measured influenza virus RNA in a hospital emergency department, finding that 46% of the RNA was associated with particles larger than 4 μm in diameter, 49% with 1–4 μm particles, and only 4% with particles smaller than 1 μm. With the same sampler, Lindsley et al. [37] measured RNA of influenza A virus and respiratory syncytial virus (RSV) in an urgent care facility. About 30% of influenza A RNA was associated with particles smaller than 1.7 μm while almost none of the RSV RNA was associated with particles of that size. Yang et al. [38] sampled aerosols in a healthcare facility, a day-care center, and an airplane using a cascade impactor and analyzed the collected particles for influenza RNA. They found that the virus was broadly distributed across particle sizes ranging from <0.25 μm to >2.5 μm in diameter; an average of 64% of the RNA was associated with particles smaller than 2.5 μm.

While data about the size distribution of particles with which viral RNA is associated are valuable, it does not provide information about whether the virus is infectious or not. Processing samples to isolate and titrate infectious virus can provide data on virus infectivity as a function of particle size. This critical information is needed to evaluate the risks that virus aerosols pose. Alonso et al. [22] infected piglets artificially with an influenza virus. Air samples were collected daily using both a wet cyclonic collector and an Andersen cascade impactor. Infectious virus was detected in only 29% of the samples in which viral RNA was found, and only for particles 2.1 μm and larger. During the outbreak of highly pathogenic avian influenza (HPAI) virus H5N2 in the U.S. Midwest in 2015, Torremorell et al. [23] sampled the air for the virus in or nearby housing for six poultry flocks using the wet cyclonic collector and two different kinds of cascade impactors. Samples were screened for influenza virus RNA using RT-PCR followed by virus isolation from all RT-PCR positive or suspect air samples. HPAI genetic material was detected inside and for up to 1,000 m from infected facilities. Infectious HPAI was isolated from air samples collected inside, immediately outside, and for up to 70 m from the infected facilities and in airborne particles larger than 2.1 μm in diameter. The Andersen cascade impactor generally measured higher concentrations of virus compared to the other samplers.

To effectively measure the sizes of particles with which infectious viruses are associated, we must identify or develop an optimal method to sample size-separated viral aerosols at a high volumetric flow rate with high efficiency to provide as low a limit of detection as possible, while keeping the viruses viable for culture-based analyses. This is the first in a series of papers in which we report the results of tests to compare a wide range of existing samplers that can collect viral aerosols using a variety of principles. The objective of our tests is to identify combinations of sampling parameters and technologies that (1) provide the greatest chance of detecting infectious virus and viral RNA and (2) most accurately measure airborne concentrations of infectious virus and viral RNA. This first paper focuses primarily on samplers that use the principles of impingement and cyclonic collection.

## Materials and methods

### Ethics statement

Protocols and procedures followed throughout the study were approved by the University of Minnesota Institutional Biosafety Committee (IBC #1808-36316H).

## Test viruses and propagation

Three different viruses were aerosolized experimentally: an avian influenza virus (AIV) subtype H9N9 as a surrogate for avian influenza viruses of pandemic potential, an H3N2 swine influenza virus (SIV) as a surrogate for human origin viruses capable of transmission between pigs and humans, and MS2 coliphage used commonly as a general virus surrogate in aerosol studies. Low-pathogenic strains of AIV and SIV were chosen as surrogates for highly pathogenic zoonotic influenza strains to which animal agriculture workers might be exposed.

The influenza viruses were grown and titrated in Madin-Darby canine kidney (MDCK) cells. The cells were grown in minimum essential medium (MEM) with Earle's Salts and L-Glutamine (Mediatech Inc., Manassas, VA, USA) supplemented with 8% fetal bovine serum (FBS), penicillin (100 U/mL), streptomycin (100 μg/mL), neomycin (90 U/ mL), and fungizone (1 μg/mL). AIV and SIV were propagated separately in MDCK monolayers at 90% confluency. The monolayers were infected after thrice washing with Hanks' balanced salt solution and maintained in MEM supplemented with 4% of 7.5% bovine serum albumin solution- fraction V (BSA; Thermo Fisher Scientific, MA), 0.65 U/mL TPCK-trypsin (Worthington biochemical Inc, NJ), and antibiotics as mentioned above. The infected cells were incubated at 37°C under 5% $CO_2$ atmosphere until cytopathic effects (CPE) appeared, usually within 2–4 days. The cultures were frozen and thawed one time followed by centrifugation at 3,000 x g for 15 minutes. The supernatants, 1 liter from each virus, were aliquoted in 45 mL amounts and stored at -80°C until used in experiments.

The MS2 coliphage (ATCC 15597- B1) was propagated and titrated in *Escherichia coli* famp (ATCC 700891) as host cells following the method described previously [39]. Briefly, 1 liter of log-phase *E. coli* culture, grown at 37°C for 5 h in 3% (w/v) tryptic soy broth (TSB, Fisher Scientific), was infected with 10 mL of virus stock and incubated overnight at 37°C with shaking at approximately 100 rpm. The virus was collected by centrifugation of the culture at 3,000 ×g for 15 min at 4°C followed by filtration through syringe filters Millex-HV25 33 mm PVDF 0.45 μm (Millipore, MA). The filtered virus stock was aliquoted in 45 mL amounts in 50 mL-Falcon centrifuge tubes and stored at -80°C until used.

## Sampling instruments

The sampling instruments, their suppliers, and their flow rates are presented in Table 1. The Andersen cascade impactor is an 8-stage impactor with a backup filter that effectively separates particles into 9 size intervals. The aerodynamic diameters for particle size separation range

**Table 1. Air samplers evaluated in tests using artificially generated viral aerosols, with the sampling airflow rate for each.**

| Sampler | Supplier (City, State) | Sampling flow rate (L/ min) |
|---|---|---|
| Non-Viable Andersen Cascade Impactor | Tisch Environmental (Cleves, OH) | 28.3 |
| Cyclonic Collector | Midwest Micro-Tek (Brookings, SD) | ~200 |
| AGI-30 impinger | Ace Glass, Inc. (Vineland, NJ) | 12.5 |
| BioSampler | SKC Inc. (Eighty Four, PA) | 12.5 |
| NIOSH Cyclone Bioaerosol Sampler | National Institute for Occupational Safety and Health (Morgantown, WV) | 3.5 |
| SpinCon II | InnovaPrep (Drexel, MO) | 450 |
| Bobcat | InnovaPrep (Drexel, MO) | 200 |
| VIVAS | University of Florida (Gainesville, FA) | 8 |

from 0.4 μm to 9.0 μm. In this study, uncoated aluminum plates were used as the collection surface for each stage except for the glass fiber filter below the final stage. The Cyclonic Collector combines cyclonic collection and impingement to capture particles. Its particle collection characteristics as a function of particle size have not been characterized well. These two instruments were included in all studies with different types of samplers to provide a basis of comparison across studies.

The AGI-30 impinger has been used commonly to sample biological particles into a liquid suspension. The BioSampler, also made from glass, utilizes cyclonic motion in addition to impingement to improve sampling efficiency [27]. Both the AGI-30 and the BioSampler have notable declines in sampling efficiency for particles smaller than 1 μm in aerodynamic diameter [40]. Like the BioSampler, the SpinCon II combines cyclonic motion with impingement, albeit at a much higher flow rate. The National Institute for Occupational Safety and Health (NIOSH) Cyclone Bioaerosol Sampler, a sampler worn by an individual, utilizes a pair of cyclones in series to collect biological aerosols into microcentrifuge tubes, followed by a backup filter [41]. Samples in the tubes can be processed directly. Using the cyclones, the sampler differentiates particles by size into fractions >4 μm, 1–4 μm, and <1 μm [36]. The Bobcat captures airborne particles on a filter which is then eluted quickly using a custom-made kit supplied by the manufacturer. The VIVAS utilizes a water-condensation growth tube to grow submicrometer particles that readily form micrometer-sized droplets that are deposited into a collection liquid by inertial impaction [42]. The temperature profile of the VIVAS sampler stages we used was 5°C at the conditioner, 40°C at the initiator, 10°C at the moderator, 20°C at the nozzle, and 5°C at the sample. To avoid cross contamination between sampling runs by VIVAS, we performed a negative sampling run for 30 min after each actual sampling run, in which we connected a HEPA filter at the instrument inlet.

The sizes and concentrations of all particles in the room were measured during tests with an optical particle counter (AeroTrak Handheld Particle Counter Model 9306, TSI Inc., Shoreview, MN). This direct-reading instrument reported particle number concentrations in six size intervals ranging from 0.3 μm to >10 μm. The instrument also recorded temperature and relative humidity in the room.

## Test setting and procedures

The tests were conducted in a room in the BSL-2 Veterinary Isolation Buildings at the University of Minnesota. These buildings are normally used for animal studies, but no animals were present at the time of these experiments. Each isolation room had an anteroom with footbaths, a sink for hand washing, and a storage area for equipment. The animal housing section of each room measured 3.24 m long x 2.34 m wide, with a height of 4.11 m; this was where viral aerosols were generated and sampled. Adjacent to the animal housing section was a 2.08 m$^2$ work area open to the animal housing. For experiments, the work area was separated from the animal housing section by two floor-to-ceiling plastic sheets that were doubled over enough to provide researchers with the ability to move from one part of the room to the other without disrupting air flow substantially. The floor of the isolation room was solid concrete and the walls were painted cinderblock. Each isolation room was ventilated separately. Air flow entered the room through a supply plenum at a flow of approximately 400 m$^3$/h.

Aliquots of stock viruses were thawed, and 45 mL suspensions were placed in a 6-jet Collison-type nebulizer (BGI Inc., Waltham, MA). To avoid foaming during nebulization process, 90 μL of antifoam Y-30 emulsion (Sigma-Aldrich, MO) were added to 45 mL virus suspension (2 μL/mL final concentration). In a preliminary experiment, the 2 μL/mL antifoam Y-30 emulsion did not affect the infectivity of the three tested viruses after 2 h exposure time at room

temperature. To compare virus recovery from samplers to as a measure of physical recovery, the virus suspensions were spiked with fluorescent tracer dye (fluorescein sodium salt, Fluka, Buchs, Switzerland) at a concentration of 1 mg/mL. As shown in Fig 1(A) and 1(B), the neb-ulizer was positioned directly in front of the opening of the supply plenum to promote mixing in the room. The nebulizer was operated at 20 psi. Preliminary tests using the optical particle counter indicated that aerosol concentrations and size distributions were uniform and stable within a few minutes in the lower half of the room opposite from the supply plenum. There-fore, samplers were positioned there as shown in Fig 1(C). Inlets to the samplers were sepa-rated to the extent possible to avoid interferences.

Samples were collected simultaneously using all samplers in each test. The nebulizer was turned on first and its operation was allowed to stabilize for 15 minutes, after which all sam-plers were turned on at the same time. All samplers were operated for 30 minutes, with the flow to the nebulizer being turned off after the samplers. The suspensions were aerosolized at a rate of approximately 0.39 mL/min during the tests. The samplers were evaluated with each of the three viruses in three replicate tests for a total of nine tests. The test order was randomized.

## Virus recovery

For virus recovery, MEM with 2% BSA, 0.2μL/mL antifoam Y-30, and the standard antibiotics was used as a collection liquid for viral aerosols of AIV and SIV. The same liquid was used for MS2 coliphage sampling but without adding antibiotics to avoid killing *E. coli* during titration. Approximately 1 liter of the same aerosol collection liquid was used to fill the inline extraction fluid container of SpinCon II, replacing the phosphate-buffered saline (PBS) solution which is the original extraction fluid supplied by the manufacturer. The SpinCon sampler was pro-grammed to aspirate approximately 10 mL of the extraction fluid container for virus recovery during each run and yielding ~4.5 mL. The Bobcat filter was eluted with a 10mL PBS foam kit provided by the manufacturer, yielding a liquid volume of 6 mL. In the Cyclonic Collector, 10 mL aliquots of the collection liquids were used, and this volume was reduced to approximately 4 mL at the end of the sampling time due to evaporation and mass transfer to the airflow. Vol-umes of 20 mL were used in both the AGI-30 impinger and the BioSampler, yielding average volumes of 7.5 mL and 15 mL, respectively. In the VIVAS sampler, we used 1.5 mL of collec-tion liquid and it increased to 2 mL at the end of the sampling time. This volume increase was attributed to the water condensation in the water-based particle growth tube of the VIVAS. For the Cyclone Bioaerosol Sampler, 3 mL of the collection liquid was added to the first stage microcentrifuge tube and 1.8 mL to the second tube to suspend the sampled virus. The final stage filter was eluted with 3 mL collection liquid. Samples from Andersen impactor plates were eluted from every plate stage using a cell scraper and 3 mL of the collection liquid. The final stage filter was eluted with 15 mL of the collection liquid. To increase the recovery of viral particles from the final stage filters of the Cyclone Biosampler and the Andersen impactor, the pH of the collection liquid was initially adjusted to pH 9 using 1M NaOH solution [41]. After rigorous vortex agitation for 3 min, the filter fibers were separated by centrifugation at $3,000 \times g$ for 10 min then the pH of the decanted supernatant containing the eluted virus was readjusted to pH 7 with 5M HCl solution. After sampling, all liquid samples were divided into 4 aliquots of ~0.5 mL in 1.5 mL sterile plastic microcentrifuge tubes and stored at -80˚C until testing.

## Virus titration

MS2 was enumerated by the plaque assay using the double agar layer (DAL) procedure. Top agar layer [0.75% Tryptic soya agar (TSA)] was autoclaved and kept molten in a 48˚C water

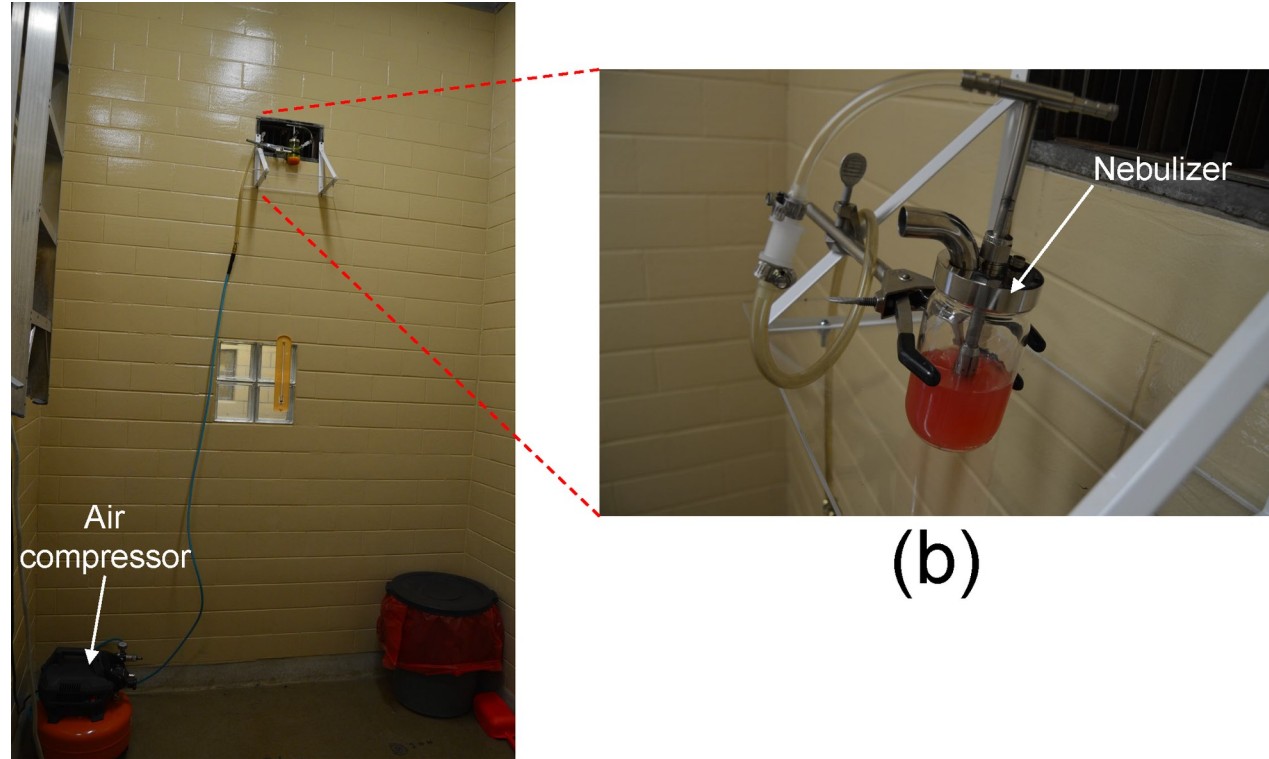

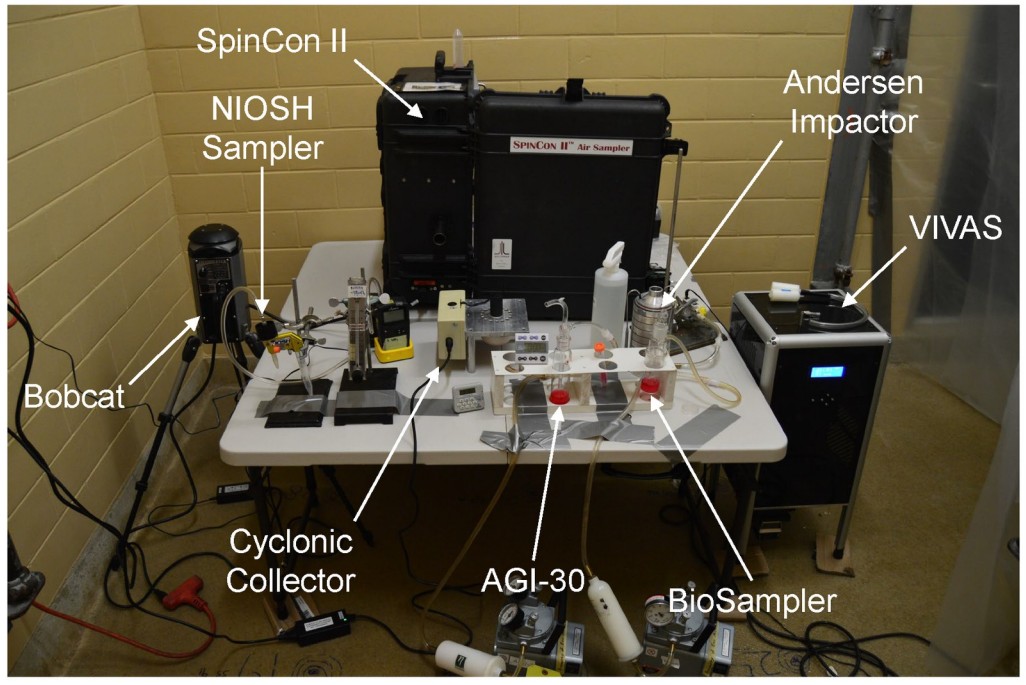

**Fig 1.** (a) Ventilation air supply plenum with (b) nebulizer placed directly in front of the opening. (c) Sampling instruments arrayed in the isolation room.

bath. Serial 10-fold dilutions ($10^0$ to $10^{-7}$) of the nebulizer suspensions and all samples were prepared in TSB. In sterile 13-mL round-bottom Falcon tubes, 100 μL of each MS2 sample dilution were added to a mixture of 5 mL top agar and 250 μL of a log-phase *E. coli* famp culture prepared in TSB, gently mixed, and poured onto 1.5% TSA bottom agar plates. After 18–24 h of incubation at 37°C, the MS2 plaques were enumerated in DAL plates, and results were expressed as PFU/mL. For any sample where there was no virus detected, a value of 0.5 PFU/mL was assigned for purposes of statistical analysis.

Infectious AIV and SIV in recovered aerosol samples were titrated in MDCK cells using the 50% tissue culture infective dose ($TCID_{50}$) method. Serial 10-fold dilutions ($10^0$ to $10^{-7}$) of the nebulizer suspensions and the samples were prepared in cell maintenance medium [(MEM with 4% BSA solution (7.5%), 0.65 U/mL TPCK-trypsin, and standard antibiotics)]. Triplicate wells of 1-day-old MDCK monolayers grown in 96-well cell culture plates were infected with 100μL aliquots of each sample dilution. Cytopathic effects (CPE) of influenza viruses was examined under inverted microscope after 5 days of incubation at 37°C, 5% $CO_2$, and humidity control. Viral titers were calculated by the Karber method [43] and expressed as $TCID_{50}$/mL.

To avoid the possible inaccurate recognition of CPE under microscope, we confirmed the positive or negative virus isolation in each well by hemagglutination (HA). For this purpose, 25 μL of the content of each well was pipetted into a corresponding well in a round-bottom 96-well microtiter plate followed by adding equal volumes (25 μL) of PBS and 0.5% turkey red blood cells (RBC) prepared in PBS. The HA assay plates were incubated at 37°C for 2 h followed by incubation at room temperature for 30 min. Wells with complete HA are recorded as positive and wells with a distinct button formation are recorded as negative.

## Molecular analysis of viral RNA

Viral RNA was extracted from 50 μL of nebulizer suspensions or samples using Ambion Mag-MAX™ -96 Viral RNA Isolation kit (Applied Biosystems by ThermoFisher Scientific, CA) according to the manufacturer's instructions on a semi-automatic King Fisher Flex Magnetic Particle Processor (Thermo Fisher Scientific, USA). The RNA was eluted in 50 μL of elution buffer in MagMAX Express-96 Deep well plates and stored at -80°C until used for viral genome quantification.

For the real-time RT-qPCR of influenza A viruses (AIV and SIV), we used the PCR primer set and probe described by Slomka et al [44]. They are specific for a region conserved in all type A influenza virus matrix genes as follows: M+25 forward primer (5' AGA TGA GTC TTC TAA CCG AGG TCG 3'), M-124 reverse primer (5'- TGC AAA AAC ATC TTC AAG TCT CTG-3'), M-124 modified reverse primer (5'- TGC AAA GAC ACT TTC CAG TCT CTG -3') and M+64 probe (5'- TCA GGC CCC CTC AAA GCC GA -3'). For the real-time RT-qPCR of MS2 coliphage, we used the PCR primer and probe set described by O'Connell et al. [45] that are specific for virus assembly protein as follows: forward primer, MS2-rRT-F (5'-GTCGCG GTAATTGGCGC-3'); reverse primer, MS2-rRT-R (5'-GGCCACGTGTTTTGATCGA-3'); and MS2-rRT-Probe (5'-AGGCGCTCCGCTACCTTGCCCT -3'). The probes and primers were manufactured by Integrated DNA Technologies (IDT Inc., IA). The reactions were performed using Ambion AgPath-ID One-Step RT-PCR kit (Applied Biosystems by ThermoFisher Scientific, CA). The reaction mixture (25 μL) for all viruses consisted of 5 μL of template RNA, 12.5 μL of 2X RT-PCR buffer, 1 μL 25X RT-PCR Enzyme Mix, 0.25 μL of 20 μM forward

primer solution (200 nM final concentration), 0.25 μL of 20 μM reverse primer solution (200 nM final concentration), 0.25 μL of 6 μM probe solution (60 nM final concentration), 1.67 μL of detection enhancer solution, and 4.08 μL of nuclease-free water. The RT-qPCR reaction mixtures were distributed in ABI MicroAmp® Fast Optical 96-Well Reaction Plates (Applied Biosystems by Thermo Fisher Scientific, CA) and loaded on Thermo Fisher Scientific Applied BioSystems QuantStudio-5 Real-Time PCR thermocycler system. The thermal cycling conditions of both AIV and SIV were 45˚C/10 min for reverse transcription (RT), 95˚C/10 min for Taq polymerase activation, and 45 PCR amplification cycles using a 94˚C/1 s denaturation step and an annealing step of 60˚C/30 s. The thermal cycling conditions of MS2 were 50˚C/30 min for RT, 95˚C/15 min for Taq polymerase activation, and 45 PCR amplification cycles using a 95˚C/15 s denaturation step and an annealing step at 55˚C/45 s. In each run of RT-qPCR, standard curve samples and no template control were used as positive and negative controls. Results were expressed as cycle threshold (Ct) values. The standard/calibration curve of AIV and SIV was constructed for absolute quantification of viral genome copy number in samples, in which serial ten-fold dilutions of a matrix gene transcript RNA with known copy number was used. For construction of MS2 standard/calibration curve, serial ten-fold dilutions of MS2-RNA extracted from a virus stock with a known titer was used. The Ct values were then used along with the standard curve to calculate the absolute genome copy number of influenza A viruses or MS2 genome copy number expressed as projected PFU. The viral genome copies or projected PFU values are referred to as total virus (infectious virus + inactivated virus) in our analyses.

## Fluorometric quantification

Fluorescein levels in the nebulizer suspension before and after sampling, as well as those from all samples, were assessed using a microplate reader (Model Synergy H1, BioTek, Winooski, VT, USA). All samples were transferred in triplicate into 96-well black plates and fluorescence intensity was measured at $\lambda = 515$ nm after excitation at $\lambda = 485$ nm. Results were expressed as fluorescein intensity per mL. A calibration curve was developed to relate the fluorescence intensity to fluorescein mass in the samples. The range of concentrations used in the calibration curve was 1–9 μg/ml. The original samples and serial ten-fold dilutions to $10^{-3}$ were measured to assure samples fell on the calibration curve. Calculated fluorescein mass was then utilized to determine fluorescein concentrations in nebulizer suspensions and total fluorescein and fluorescein air concentrations in samples.

## Data analysis

To evaluate the impact that nebulization had on virus infectivity while controlling for any changes in concentration of the suspending liquid due to evaporation, the infectious virus and fluorescein concentrations in the nebulizer suspensions before and after each 30-min test were used to calculate the quantity γ [28] as

$$\gamma = \frac{\left(\frac{c_V}{c_F}\right)_a}{\left(\frac{c_V}{c_F}\right)_b} \tag{1}$$

in which $c_V$ is the infectious virus concentration, $c_F$ is the fluorescein concentration, and subscripts b and a indicate before and after the test, respectively. When γ = 1, the ratio of the infectious virus concentration to the fluorescein concentration has remained the same and the virus in the suspension has not been inactivated by repeatedly passing through the nebulizer spray nozzle. γ was evaluated for all three tests with each virus, and geometric means and 95%

confidence intervals were calculated from these data. Using Microsoft Excel, t-tests were run for each virus to determine if the geometric mean of γ was different from 1.

The amount of infectious virus, total virus, and fluorescein collected by each sampler were calculated by multiplying the concentrations of infectious virus, viral RNA, and fluorescein determined from virus titration, molecular analyses, and fluorometry by the total volume of liquid in which the sampled virus was recovered. Airborne concentrations of infectious virus, viral RNA, and fluorescein measured by each sampler were calculated by dividing the amounts sampled by the volume of air sampled during the 30-min test. To normalize residuals as a function of concentration, concentrations were log-transformed for statistical analyses. Geometric means and geometric standard deviations were calculated for each sampler for amounts of infectious virus and viral RNA sampled and for air concentrations for both parameters. To determine if infectious virus and viral RNA collected and measured infectious virus and viral RNA air concentrations differed significantly as a function of sampler, one-way analyses of variance were conducted using SAS (Version 9.4, Cary, NC). Multiple comparisons of geometric means were performed according to Tukey's method. For all comparisons, a p-value ≤0.05 was considered significant.

The relative recovery rate of virus from the sample, $R_{rel}$, indicates the fraction of nebulized virus that remains infectious after sampling and analysis [28,46,47]. If all of the virus in a sample is recovered infectious without virus inactivation in the air or after sampling and without experimental error, $R_{rel} = 1$. Relative recovery can be calculated as

$$R_{rel} = \frac{\left(\frac{c_V}{c_F}\right)_s}{\left(\frac{c_V}{c_F}\right)_n} \tag{2}$$

in which subscripts s and n indicate assessments of infectious virus and fluorescein concentrations in the samples and in nebulizer suspensions, respectively. Concentrations in the nebulizer suspensions were calculated as the geometric means of the concentrations before and after the test. Geometric means and geometric standard deviations of $R_{rel}$ were calculated from the three replicates for each virus. To determine if $R_{rel}$ differed significantly as a function of sampler, one-way analyses of variance were conducted using SAS. Multiple comparisons of geometric means were performed according to Tukey's method. A p-value ≤ 0.05 was considered significant.

Number concentration data from the optical particle counter in each size interval were converted to mass concentrations assuming that particles were spherical with a diameter equal to the geometric midpoint of the size interval and that the density of the particles was 1.0 g/cm$^3$. The mass concentration in each size interval was then converted to a fraction of the total mass concentration, and the fraction was normalized by dividing by the width of the interval on a logarithmic scale. Means and standard deviations for the normalized mass fractions were then calculated for each virus in each size interval.

Infectious virus, total virus, and fluorescein concentrations from the various stages of the Andersen cascade impactor and the Cyclone Bioaerosol Sampler were divided by the total concentrations across all stages to compute fractions in each interval. The fractions were normalized by dividing by the width of the size interval on a logarithmic scale. For each virus, means and standard deviations were then calculated for each size interval.

## Results

All data collected and used in this study are provided in S1 Dataset. The average temperature across the nine tests was 27.7˚C with a standard deviation of 2.1˚C. The average relative

humidity was 44.3% with a standard deviation of 6.4%. The geometric mean (upper 95% confidence limit, lower 95% confidence limit) of γ was 1.4 (0.32,6.3) for MS2, 2.1 (0.24,18) for SIV, and 0.83 (0.044,16) for AIV. While gamma varied substantially, we can conclude that none of the viruses were inactivated systematically by the nebulization process during the test period because the geometric means of γ were not significantly different from 1.

Average MS2 concentrations in the nebulizer suspensions were $9.89 \times 10^7$ PFU/mL and $3.05 \times 10^8$ projected PFU/mL for infectious and total virus, respectively. For SIV, average concentrations in the suspensions were $4.62 \times 10^6$ TCID$_{50}$/mL and $4.96 \times 10^9$ genome copies/mL for infectious and total virus, respectively. Average AIV concentrations in the nebulizer were $3.14 \times 10^4$ TCID$_{50}$/mL and $3.04 \times 10^{10}$ genome copies /mL for infectious and total virus, respectively.

Fig 2 shows geometric means of the quantity of infectious virus and viral RNA collected by each of the air samplers for MS2, SIV, and AIV. Error bars represent one geometric standard deviation on either side of the geometric mean. Letters at the bottom of each bar indicate samplers that can or cannot be distinguished from each other statistically. Samplers having the same letter are statistically indistinguishable; those with no letter in common are significantly different.

Fig 3 presents the geometric means of the airborne concentrations of infectious virus and viral RNA for each air sampler for each of the three viruses. With our knowledge of ventilation air flow entering the room (Q), the generation rate of aerosol from the nebulizer suspension (G), and the virus concentrations in the nebulizer suspension (c$_s$), we can estimate the expected concentration of virus in the air (c$_a$), assuming that the air in the room is well-mixed, using the equation

$$c_a = \frac{Gc_s}{Q}. \tag{3}$$

With values of G, c$_s$, and Q reported above, we estimated infectious virus air concentrations of $5.77 \times 10^6$ PFU/m$^3$, $2.70 \times 10^5$ TCID$_{50}$/m$^3$, and $1.84 \times 10^3$ TCID$_{50}$/m$^3$ for MS2, SIV, and AIV, respectively, if none of the viruses were inactivated after nebulization. For total virus from molecular analyses, we estimated air concentrations of $1.78 \times 10^7$ projected PFU/m$^3$, $2.89 \times 10^8$ viral genome copy/m$^3$, and $1.77 \times 10^9$ viral genome copy /m$^3$ for MS2, SIV, and AIV, respectively. Statistical analyses indicate that the concentrations shown in Fig 3 are all either lower than or indistinguishable from these estimated concentrations.

Relative recoveries for MS2, SIV, and AIV for each of the air samplers are presented in Fig 4, with error bars representing one geometric standard deviation on either side of the geometric mean. Using t tests of significance with p = 0.05, none of the recoveries for MS2 or AIV were significantly different from 1, meaning loss of virus infectivity could not be demonstrated for these viruses relative to infectivity in the nebulizer suspension. For SIV, however, relative recoveries for the Andersen impactor, BioSampler, VIVAS, SpinCon II, and NIOSH samplers were all significantly lower than 1, meaning that some of the SIV was inactivated while airborne or during sampling.

Fig 5 shows mean fractional particle size distributions normalized by the width of the size interval measured during tests with MS2. Data from the optical particle counter were used to calculate size distribution by mass for the total aerosol. Size distributions are also presented for infectious virus, viral RNA, and fluorescein mass concentrations sampled with the Andersen impactor and the NIOSH sampler. In Fig 6, size distributions are presented with standard deviations for tests with all three viruses. This figure only includes data from the optical particle counter and the viral RNA analyses from the Andersen impactor and the NIOSH sampler.

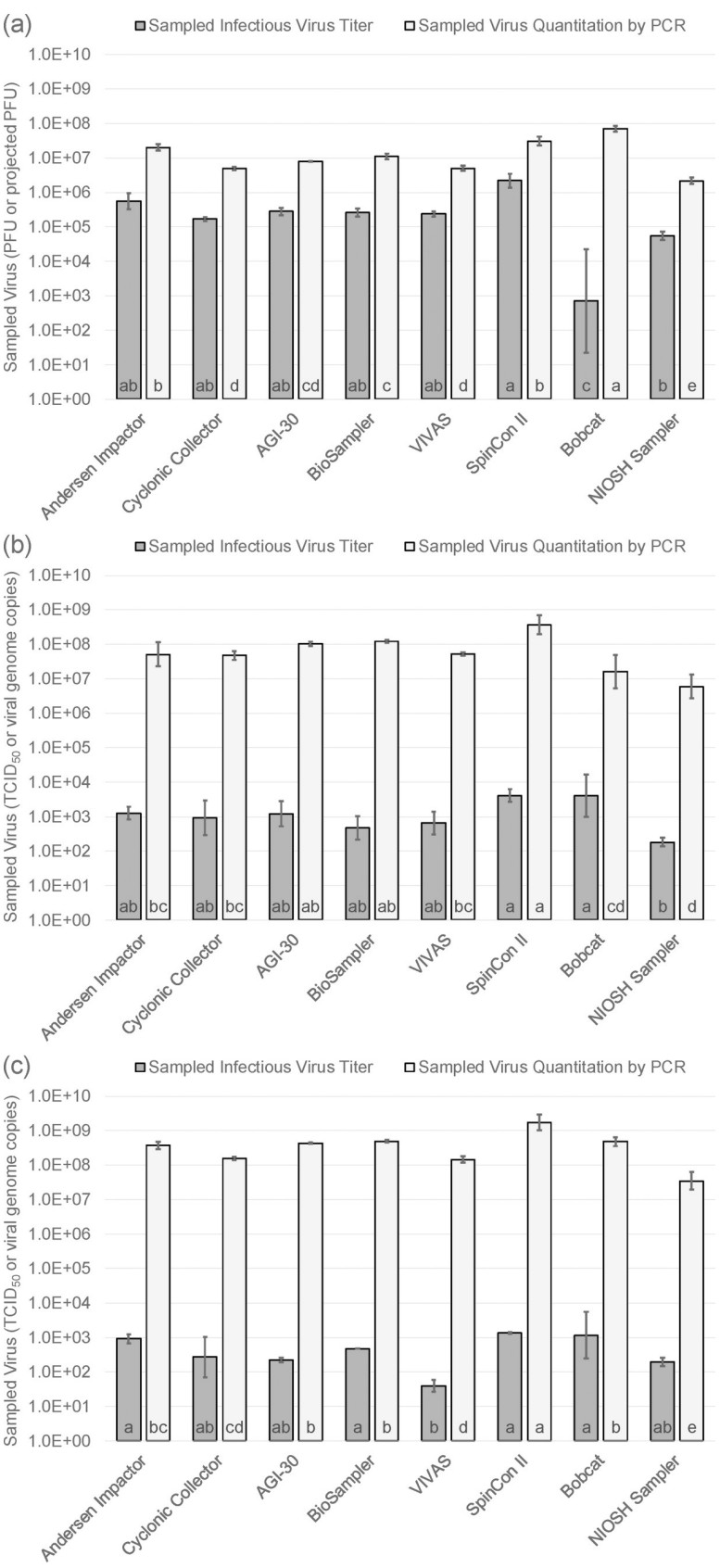

**Fig 2.** Sampled infectious virus and viral RNA for air samplers as measured for (a) MS2 bacteriophage, (b) H3N2 swine influenza virus, and (c) H9N9 avian influenza virus. Bars represent geometric means and error bars represent ± one geometric standard deviation. Same letters indicate geometric means that are not significantly different from one another.

## Discussion

Not surprisingly, samplers that draw in more air generally capture more virus, as shown in Fig 2. Among the eight samplers, the SpinCon II collected either the highest or second highest amount of viral RNA for each of the test viruses. The Bobcat recovered significantly more MS2 RNA than any of the other samplers followed by the SpinCon II and the Andersen impactor. The SpinCon II, AGI-30 impinger, and BioSampler were in the highest grouping for recovering SIV RNA, while the SpinCon II was highest for AIV RNA recovery followed by the Andersen impactor, AGI-30, BioSampler, and Bobcat in the next grouping. The SpinCon II recovered the most infectious virus for all three viruses, although this recovery was not significantly different than recovery by several other samplers for any of the viruses.

The NIOSH sampler, which had the lowest flow rate of any of the samplers, recovered the least viral RNA for all three viruses. The Bobcat had the lowest infectious virus recovery for MS2. The reason for this is unclear, but it may be that MS2 is more difficult to elute from the Bobcat filter element than the influenza viruses because it is a smaller virus and may be attached more strongly to filter fibers. The finding that lower flow rate samplers generally collected less RNA and lower titers of infectious virus is not surprising given that they are processing less air. An interesting set of experiments would be to run the samplers so that they process equal volumes of air and then determine the titers from each device.

A high flow rate sampler like the SpinCon II that seems to recover both infectious and total virus effectively may be the best option for detecting virus in a workplace when concentrations are likely to be low. However, having a high flow rate does not by itself make a sampler worthy. The Cyclonic Collector, with a sampling flow of about 200 L/min, generally recovered less infectious and total virus than other high flow rate samplers and even less than some samplers with lower flow rates.

The results in Fig 3 suggest that, while high flow rate samplers may be better for detecting infectious virus and viral RNA in the air, airborne virus concentrations are measured more accurately by lower flow rate samplers. For all three viruses, the three highest flow rate samplers–the SpinCon II, Bobcat, and Cyclonic Collector–were never in the highest grouping for measurement of airborne RNA concentrations. In addition, they tended to measure lower concentrations of infectious virus, although the differences were not always significant from samplers measuring higher concentrations. On the other hand, the lower flow rate samplers–the Andersen impactor, AGI-30, BioSampler, VIVAS, and NIOSH samplers–were always in the highest or second highest grouping of samplers for measuring infectious and total airborne virus concentrations for all three viruses. Notably, all five of these samplers, but especially the AGI-30 and the BioSampler, measured viral RNA concentrations that were close to expected concentrations based on the well-mixed room assumption.

The reason that lower flow rate samplers seem to measure airborne virus concentrations more accurately is uncertain. One feature of the higher flow rate samplers is that they consolidate a sample from a large amount of air into a similar volume of liquid as the lower flow rate samplers. It is possible that these consolidation processes damage the virus RNA in some way that reduces the measured infectious and total virus concentrations. In addition, the high flow rate samplers showed more evaporation of the collection medium based on visual observations. An increase in salts concentration of the collection medium due to evaporation might

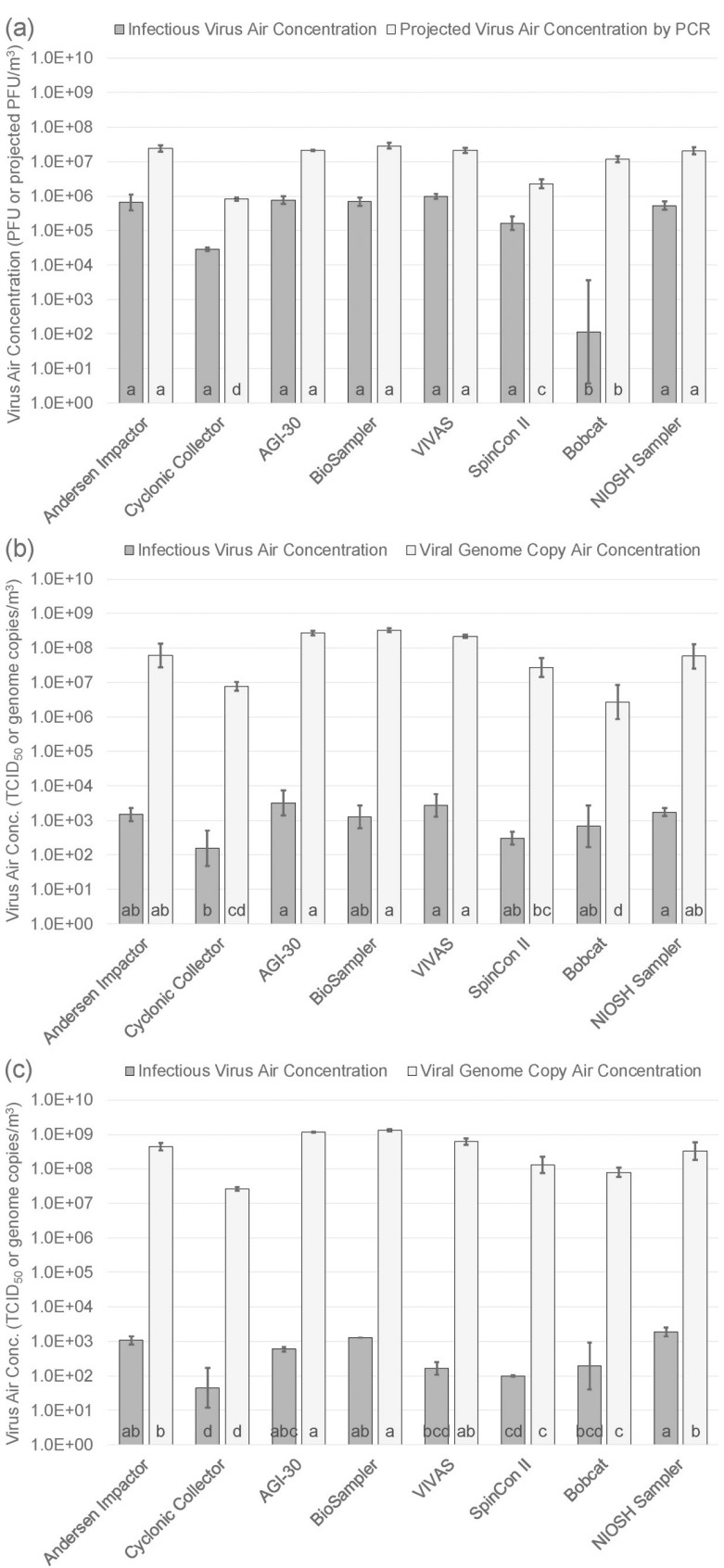

**Fig 3.** Infectious virus and viral RNA air concentrations measured by samplers for (a) MS2 bacteriophage, (b) H3N2 swine influenza virus, and (c) H9N9 avian influenza virus. Bars represent geometric means and error bars represent ± one geometric standard deviation. Same letters indicate geometric means that are not significantly different from one another.

contribute to reductions in the infectivity of the collected viral particles, as this has been demonstrated with other viruses [48].

Another possible reason for reduced concentration measurements in high flow rate sampler is that the process of sampling at high flow rates may result in lower removal or aspiration efficiencies for some particle sizes. For example, unpublished measurements made by the authors for the Cyclonic Collector suggest that particles smaller than 2 μm and larger than 8 μm may be removed at less than 100% efficiency. The low efficiency for small particles may be due to insufficient inertia whereas the low efficiency for larger particles may be related to particle bounce on interior surfaces. In addition, it is not clear that the inlets for any of the high flow rate samplers were designed to maximize aspiration efficiency for large particles.

Another potential explanation is that the particle-free air exhausted from the high flow rate samplers might dilute the viral aerosols entering the samplers. However, the exhaust air from these samplers was directed away from sampling inlets when possible. The exhaust air would also be expected to dilute air entering adjacent samplers given tight spacing, but this was not

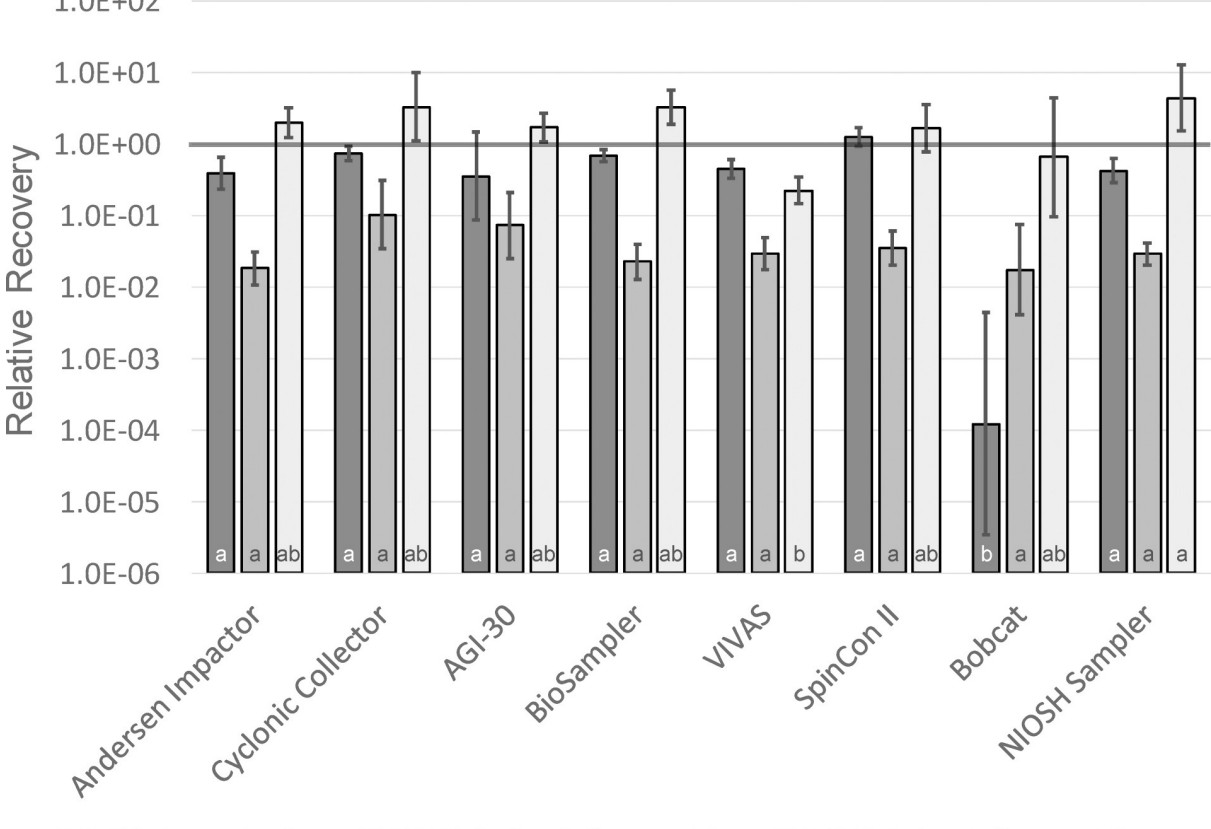

**Fig 4. Relative recovery measured for MS2 bacteriophage, H3N2 swine influenza virus, and H9N9 avian influenza virus for each of the air samplers.** Bars represent geometric means and error bars represent ± one geometric standard deviation. Same letters indicate geometric means that are not significantly different from one another within each type of virus.

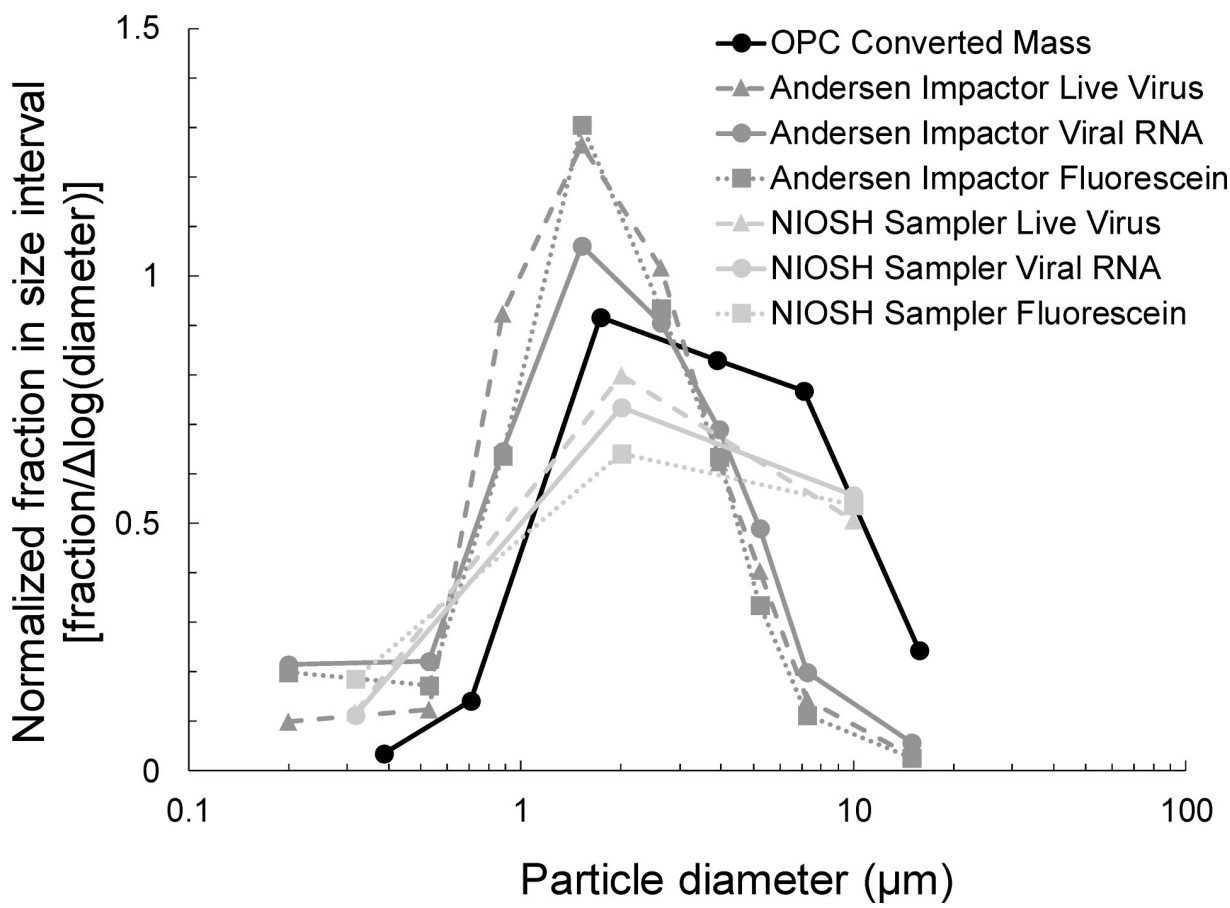

**Fig 5. Mean fractional size distributions normalized by the width of the size interval on a logarithmic scale for the size distribution by mass calculated from optical particle counter data and for live virus, viral RNA, and fluorescein sampled with the Andersen cascade impactor and the NIOSH Cyclone Bioaerosol Sampler during tests with MS2 bacteriophage.**

observed. Therefore, this is not a likely reason for the lower concentrations found with the high flow rate samplers.

Relative recoveries were generally close to 1 for all of the samplers for MS2 and AIV, as shown in Fig 4. In fact, none of these relative recoveries were significantly different from 1 according to statistical analyses. The relative recoveries for SIV for several of the samplers were significantly lower than 1. The reason for these differences among the viruses is not clear. It is possible that SIV is more susceptible to inactivation while airborne and after sampling than the other viruses.

Figs 5 and 6 indicate that the size distributions measured with the Andersen impactor were generally shifted to smaller particle sizes than those measured using the optical particle counter and the NIOSH sampler. if the sampling methods have no effect on the measurements, the size distributions from all three instruments should be similar because virus concentrations should be proportional to particle mass concentrations as a function of size. However, the observation that the Andersen impactor measures distributions shifted to smaller sizes may be a reflection of particle bounce that has been documented previously for cascade impactors [49]. A way to avoid particle bounce is to coat impactor stages with a material that prevents bounce, such as grease [50–52]. However, grease would likely interfere with virus recovery and analysis and is not a practical option. A potential solution may be to draw from the pharmaceutical aerosol

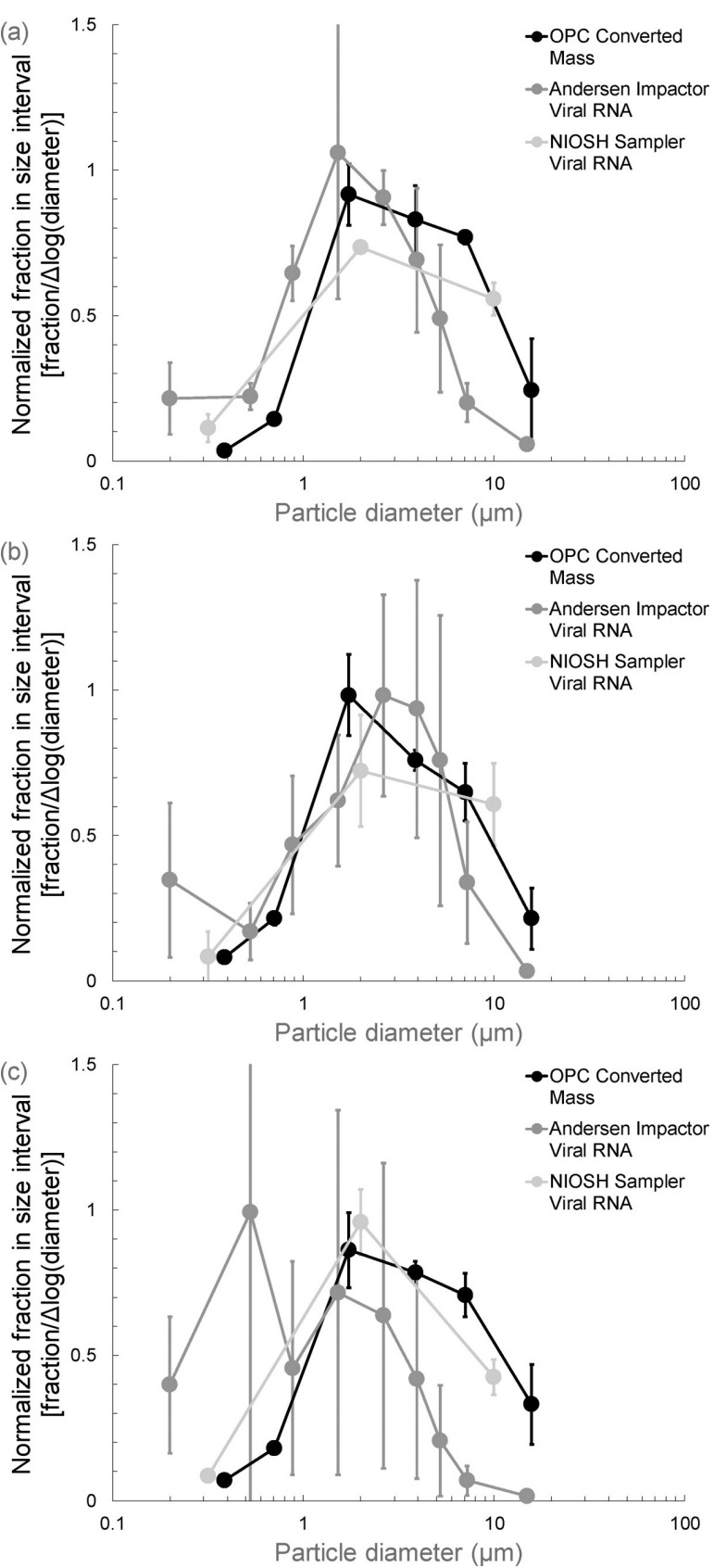

**Fig 6.** Mean fractional size distributions normalized by the width of the size interval on a logarithmic scale for the size distribution by mass calculated from optical particle counter data and for viral RNA sampled with the Andersen cascade impactor and the NIOSH Cyclone Bioaerosol Sampler for (a) MS2 bacteriophage, (b) H3N2 swine influenza virus, and (c) H9N9 avian influenza virus. Bars represent ± one standard deviation from the mean value.

literature in which researchers have used water-soluble coatings on impactor stages to prevent particle bounce [53–55]. These coatings may interfere less than grease with virus recovery.

Portability and ease of use are important considerations in selecting any sampler. The Bobcat was the easiest of the eight samplers to use while also being easily portable. The AGI-30 impinger, BioSampler, and NIOSH sampler are also relatively easy to use. The Andersen impactor, SpinCon II, and VIVAS samplers were all more complicated, making practical use in the field more difficult. In addition, ease of use cannot be the most important consideration. The Cyclonic Collector is extremely easy to use, for example, but measurements show that it recovers virus poorly and measures air concentrations that are lower than those observed with other samplers.

The measurements made in this research have several limitations. We were only able to include six impinger/cyclone samplers in addition to the Andersen impactor and the Bobcat; many others are available. We did not have the ability to tightly control the temperature and relative humidity in the test facility. Although environmental conditions did not vary widely, they were not the same for all tests. All virus aerosols were generated artificially from nebulizer suspensions. Therefore, they do not match the complexity of viral aerosols that occur naturally. We also did not test the samplers over a range of concentrations. Testing the samplers at concentrations more like those found in real workplace settings may give a better indication of the ability of samplers to detect low concentrations of virus. Additional tests comparing the samplers in field settings would offer opportunities to address these last two limitations. While the samplers were in a section of the room that had relatively stable and uniform particle concentrations, there may have been small differences in concentrations to which each sampler was exposed during tests.

## Conclusions

Tests were conducted to compare the ability of six impinger/cyclone air samplers, a filter-based sampler, and a cascade impactor to collect airborne viruses in an experimental setting. In general, higher quantities of virus were recovered by high flow rate samplers compared to the low flow samplers. However, lower flow rate samplers were able to measure higher, and likely more accurate, infectious virus and viral RNA air concentrations than the high flow samplers. To detect and accurately assess airborne viruses in animal agriculture and other settings, a two-sampler approach may be warranted. A high flow sampler is likely to provide low limits of detection to determine if virus is present in the air. If virus is detected, a lower flow sampler might then be used to more accurately measure airborne virus concentrations.

## Supporting information

**S1 Dataset. This Microsoft Excel file contains all of the data collected and used in this study.**
(XLSX)

## Acknowledgments

The authors thank InnovaPrep LLC for their loan of the SpinCon II and Bobcat instruments utilized in this research, the University of Florida and Aerosol Devices for loaning the VIVAS,

and the National Institute for Occupational Safety and Health for its loan of the Cyclone Bioaerosol Sampler.

## Author Contributions

**Conceptualization:** Peter C. Raynor, Montserrat Torremorell, Sagar M. Goyal.

**Data curation:** Peter C. Raynor.

**Formal analysis:** Peter C. Raynor, Adepeju Adesina, Hamada A. Aboubakr.

**Funding acquisition:** Peter C. Raynor.

**Investigation:** Adepeju Adesina, Hamada A. Aboubakr, My Yang.

**Methodology:** Adepeju Adesina, Hamada A. Aboubakr, My Yang, Montserrat Torremorell, Sagar M. Goyal.

**Project administration:** Peter C. Raynor.

**Supervision:** Peter C. Raynor, Montserrat Torremorell, Sagar M. Goyal.

**Writing – original draft:** Peter C. Raynor.

**Writing – review & editing:** Adepeju Adesina, Hamada A. Aboubakr, My Yang, Montserrat Torremorell, Sagar M. Goyal.

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
