## [Decision Letter · Decision Letter 0]

6 Nov 2020

PONE-D-20-30871

Comparison of samplers collecting airborne influenza viruses: 1. Primarily impingers and cyclones

PLOS ONE

Dear Dr. Raynor,

Thank you for submitting your manuscript to PLOS ONE. After careful consideration, we feel that it has merit but does not fully meet PLOS ONE’s publication criteria as it currently stands. Therefore, we invite you to submit a revised version of the manuscript that addresses the points raised during the review process. All suggestions and questions of the reviewers need to be addressed for acceptance of the manuscript for publication.

We look forward to receiving your revised manuscript.

Kind regards,

Sander Herfst

Academic Editor

PLOS ONE

Journal Requirements:

Reviewers' comments:

**Comments to the Author**

1. Is the manuscript technically sound, and do the data support the conclusions?

Reviewer #1: Yes

Reviewer #2: Yes

2. Has the statistical analysis been performed appropriately and rigorously? 

Reviewer #1: Yes

Reviewer #2: Yes

3. Have the authors made all data underlying the findings in their manuscript fully available?

Reviewer #1: Yes

Reviewer #2: Yes

4. Is the manuscript presented in an intelligible fashion and written in standard English?

Reviewer #1: Yes

Reviewer #2: Yes

5. Review Comments to the Author

Reviewer #1: The paper compares the performance of eight aerosol samplers when collecting a bacteriophage and two influenza viruses. I have minor comments and questions.

When comparing samplers in a test chamber, a constant concern is always that the samplers are sampling aerosols with the same quantity and particle size distribution, since this directly affects the results. Steps taken often include placing the samplers on a rotating table, moving the samplers among the different test locations for different experiments, and/or adding fans to a test chamber to mix the air. In this study, the investigators did use a particle counter to check for significant variations in the particle concentration at different locations, but in the future they should consider taking additional steps and performing additional experiments to help ensure that the results they are seeing are due to actual differences between the samplers and not due to differences in aerosol concentrations.

The investigators are using fluorescein in their experiments as a tracer, which is common in aerosols experiments. The fluorescence of fluorescein is pH sensitive. Since they did all their experiments using the same buffer, their pH was probably consistent, but it is a factor to consider in the future.

Kesavan, J and RW Doherty (2000). Use of Fluorescein in Aerosol Studies. Report No. ECBC-TR-103. Aberdeen Proving Ground, MD: US Army Edgewood Chemical Biological Center, 2000, 23 p. Available at https://apps.dtic.mil/dtic/tr/fulltext/u2/a384058.pdf. Accessed April 3, 2019.

In Table 1, it would be helpful to include the cut-off diameters for the different samplers. The AGI-30 in particular is not very effective for particles below about 1 micrometer. I don’t know what the cut-off diameter is for the cyclonic collector, but if it is higher, that may explain the lower performance reported in the study.

The results indicate that the gamma values for MS2 and SIV were somewhat greater than 1. If my understanding is correct, theoretically 1 should be the maximum value. Do the investigators have any thoughts on why this occurred?

In the discussion, the investigators say, “The results in Fig 3 suggest that, while high flow rate samplers may be better for detecting infectious virus and viral RNA in the air, airborne virus concentrations are measured more accurately by lower flow rate samplers.” They discuss a few possible explanations for this. Are there other possibilities that should be considered? Are the aspiration efficiencies lower for the higher flow rate samplers? Are they recirculating filtered air from the outlet back into the inlet?

Later in the discussion, the investigators say “However, the observation that the Andersen impactor measures distributions shifted to smaller sizes may be a reflection of particle bounce that has been documented previously for cascade impactors.[48] A way to avoid particle bounce is to coat impactor stages with a material that prevents bounce, such as grease.[49-51] However, that procedure would likely interfere with virus recovery and analysis and hence is not a practical option.” The pharmaceutical industry routinely uses water-soluble coatings for impactor stages which may be more practical than grease:

Mitchell, JP (2003). Practices of Coating Collection Surfaces of Cascade Impactors: A Survey of Members of the European Pharmaceutical Aerosol Group (EPAG). Drug Delivery to the Lungs – XIV, December 11-12, 2003, London, UK, The Aerosol Society. https://www.researchgate.net/publication/288993318

Mitchell, JP, MW Nagel, V Avvakoumova, H MacKay and R Ali (2009). The abbreviated impactor measurement (AIM) concept: part 1--Influence of particle bounce and re-entrainment-evaluation with a "dry" pressurized metered dose inhaler (pMDI)-based formulation. AAPS PharmSciTech 10(1): 243-51. https://www.ncbi.nlm.nih.gov/pubmed/19280348

Khalili, SF, S Ghanbarzadeh, A Nokhodchi and H Hamishehkar (2018). The effect of different coating materials on the prevention of powder bounce in the next generation impactor. Res Pharm Sci 13(3): 283-287. https://www.ncbi.nlm.nih.gov/pubmed/29853937

In the acknowledgements, NIOSH is the National Institute for Occupational Safety and Health (“for”, not “of”).

Reviewer #2: The authors of this manuscript compared the performance of various low and high flow air samplers for the collection of MS2, an avian and a swine influenza virus. For each air sampler the amount of collected virus RNA and infectious virus was determined and expressed as 1) total collection of virus after 30 minutes of sampling and 2) per cubic meter. In addition, the loss of virus infectivity due to the nebulization process was determined, as well as the relative recovery rate of the viruses after air sampling. The size distribution for the different viruses collected by the Andersen Cascade Impactor and the NIOSH sampler was determined as well. The authors concluded from these experiments that high flow samplers in general collected more virus, while low flow rate samplers collected virus amounts more accurately and therefore suggest to use a combination of both kind of air samplers in field settings.

General remark about the experimental set-up:

All air samplers run simultaneously for 30 minutes. This would suggest that a high rate air sampler such as the SpinConII with a flow rate of 450 LPM would sample 13500 L in 30 minutes, compared to a low flow sampler like the NIOSH that samples only 105 L in 30 minutes with a flow rate of 3.5 LPM. Therefore, it is not surprising that within the same amount of time a high flow sampler might collect more virus. In addition, this also suggests that a low flow sampler is in competition with the strong air flow of high flow samplers. To get a more accurate impression of the performance of each sampler, the authors could have tested the air samplers separately and let the samplers draw the same amount of air (i.e. shorter or longer running time for either high or low flow samplers). This is a limitation of the current set-up that should be noted in the discussion.

Collection medium Andersen Impactor

The authors should provide more information on the collection medium used in the stages of the impactor. The performance of the impactor can be influenced by the collection medium used, also with regard to the shift in size distribution as it was observed in the present study.

Overall this is a nice study that compared a wide range of air samplers for the collection of aerosolized viruses.

6. PLOS authors have the option to publish the peer review history of their article (what does this mean?). If published, this will include your full peer review and any attached files.

Reviewer #1: No

Reviewer #2: No

---

## [Author Response · Author response to Decision Letter 0]

16 Dec 2020

Comments from Academic Editor

Comment 1: Please ensure that your manuscript meets PLOS ONE's style requirements, including those for file naming. The PLOS ONE style templates can be found at

Response 1: Thank you for pointing out our inattention to the style requirements. We have renamed our figure and supporting information files according to the requirements. In addition, we have edited our headings and double-spaced our references to meet the style requirements. 

Comment 2: Please include captions for your Supporting Information files at the end of your manuscript, and update any in-text citations to match accordingly. Please see our Supporting Information guidelines for more information: http://journals.plos.org/plosone/s/supporting-information.

Response 2: Thank you for pointing out that we did not provide a caption for our supporting information. We added a caption on the final page of the revised manuscript. We also added an in-text citation to the supporting information dataset file at the very beginning of the Results section on p. 17.

Comments from Reviewer #1 

Comment 3: When comparing samplers in a test chamber, a constant concern is always that the samplers are sampling aerosols with the same quantity and particle size distribution, since this directly affects the results. Steps taken often include placing the samplers on a rotating table, moving the samplers among the different test locations for different experiments, and/or adding fans to a test chamber to mix the air. In this study, the investigators did use a particle counter to check for significant variations in the particle concentration at different locations, but in the future they should consider taking additional steps and performing additional experiments to help ensure that the results they are seeing are due to actual differences between the samplers and not due to differences in aerosol concentrations.

Response 3: We agree with the reviewer's comment that uniformity of the aerosol is a significant concern in tests like these. A rotating table was not a practical option due to the size, sensitive nature, and AC power requirements of some of the instruments. Our team is reluctant to use fans for mixing because our tests have shown that fans can remove some of the aerosol from the test space. Randomly assigning locations for the instruments was somewhat feasible, although the test space is small enough that this would have presented significant challenges. The isolation room building was torn down a few months after this study in favor of a replacement with somewhat larger test spaces. This may make repositioning samplers randomly for each experiment more feasible in future studies. 

Despite not taking the approaches suggested by the reviewer, we believe that the data show that the aerosol was uniform. Figure 3 indicates that five of the samplers yielded quite similar viral RNA concentrations for all three viruses even though they were not all placed adjacent to one another. 

Comment 4: The investigators are using fluorescein in their experiments as a tracer, which is common in aerosols experiments. The fluorescence of fluorescein is pH sensitive. Since they did all their experiments using the same buffer, their pH was probably consistent, but it is a factor to consider in the future.

Kesavan, J and RW Doherty (2000). Use of Fluorescein in Aerosol Studies. Report No. ECBC-TR-103. Aberdeen Proving Ground, MD: US Army Edgewood Chemical Biological Center, 2000, 23 p. Available at https://apps.dtic.mil/dtic/tr/fulltext/u2/a384058.pdf. Accessed April 3, 2019.

Response 4: The reviewer is correct. We should have adjusted the pH upward prior to the fluorescence measurements. We have done this in previous work; it was an oversight not to have made the adjustment in this study. However, the buffer is stable. With our calibration, therefore, the readings should still be accurate. 

Comment 5: In Table 1, it would be helpful to include the cut-off diameters for the different samplers. The AGI-30 in particular is not very effective for particles below about 1 micrometer. I don’t know what the cut-off diameter is for the cyclonic collector, but if it is higher, that may explain the lower performance reported in the study.

Response 5: We would agree with the reviewer's comment if data were available on cut-off diameters for all of the instruments. We have already indicated in the text on p. 9 that the AGI-30 and the BioSampler have significant losses in sampling efficiency for particles smaller than 1 μm. The size intervals for the Andersen impactor and the NIOSH sampler were provided on pp. 8 and 9, and we indicated on p. 9 that the VIVAS works by growing submicrometer particles into ones that can more easily be collected. The sampling efficiency by particle size for the SpinCon II, the Bobcat, and the Cyclonic Collector have not been thoroughly characterized. We believe that we have reported the size dependent properties as thoroughly as possible in the text, and that adding data for only some of the samplers in Table 1 would not be helpful. Therefore, we have not revised the table.

We have taken some very rough measurements regarding the sampling efficiency of the Cyclonic Collector, which indicate that it may not sample at 100% efficiency for particles smaller than 2 μm and larger than 8 μm. We have referred to this in the Discussion on p. 22.

Comment 6: The results indicate that the gamma values for MS2 and SIV were somewhat greater than 1. If my understanding is correct, theoretically 1 should be the maximum value. Do the investigators have any thoughts on why this occurred?

Response 6: The main reason that we calculate gamma is to determine if a substantial proportion of a virus is being inactivated by the nebulization process. If we were to do this for vegetative bacteria, for example, gamma values would be close to 0. In theory, the measured gamma could be greater than or less than 1. It would be less than 1 if the virus were inactivated by repeatedly being passed through the nozzle in the nebulizer. On the other hand, gamma could be greater than 1 if the virus is (a) not inactivated and (b) is also not suspended evenly over time in the nebulizer fluid so that proportionally more fluorescein would be nebulized than virus. 

Whether the geometric mean values are greater than or less than 1 is not very important here. As we indicate, the population of gamma values in our tests cannot be distinguished statistically from 1 for any of the viruses. This means that the viruses are not being systematically inactivated over the course of each test period, nor is the virus suspension systematically becoming non-homogeneous. The geometric means differ from 1 and the 95% confidence limits around gamma for each of the viruses are fairly large due to the inherent variability of working with viruses. In addition, a sample size of three for each virus is not ideal. With more samples, we would have geometric means closer to 1 and tighter error bars, as in some of our previous work. 

We have chosen not to make revisions to the manuscript around this point because we believe these explanations are tangential to the main point, which is that we do not see systematic inactivation of the virus due to nebulization.

Comment 7: In the discussion, the investigators say, “The results in Fig 3 suggest that, while high flow rate samplers may be better for detecting infectious virus and viral RNA in the air, airborne virus concentrations are measured more accurately by lower flow rate samplers.” They discuss a few possible explanations for this. Are there other possibilities that should be considered? Are the aspiration efficiencies lower for the higher flow rate samplers? Are they recirculating filtered air from the outlet back into the inlet?

Response 7: The reviewer makes an excellent point. We were a little limited in indicating possible reasons for the high flow rate samplers yielding lower concentrations. In response, we added two paragraphs in the Discussion section on pp. 22-23, one addressing removal and aspiration efficiency and the other addressing the possible effects of sampler exhaust. 

Comment 8: Later in the discussion, the investigators say “However, the observation that the Andersen impactor measures distributions shifted to smaller sizes may be a reflection of particle bounce that has been documented previously for cascade impactors.[48] A way to avoid particle bounce is to coat impactor stages with a material that prevents bounce, such as grease.[49-51] However, that procedure would likely interfere with virus recovery and analysis and hence is not a practical option.” The pharmaceutical industry routinely uses water-soluble coatings for impactor stages which may be more practical than grease:

Mitchell, JP (2003). Practices of Coating Collection Surfaces of Cascade Impactors: A Survey of Members of the European Pharmaceutical Aerosol Group (EPAG). Drug Delivery to the Lungs – XIV, December 11-12, 2003, London, UK, The Aerosol Society. https://www.researchgate.net/publication/288993318

Mitchell, JP, MW Nagel, V Avvakoumova, H MacKay and R Ali (2009). The abbreviated impactor measurement (AIM) concept: part 1--Influence of particle bounce and re-entrainment-evaluation with a "dry" pressurized metered dose inhaler (pMDI)-based formulation. AAPS PharmSciTech 10(1): 243-51. https://www.ncbi.nlm.nih.gov/pubmed/19280348

Khalili, SF, S Ghanbarzadeh, A Nokhodchi and H Hamishehkar (2018). The effect of different coating materials on the prevention of powder bounce in the next generation impactor. Res Pharm Sci 13(3): 283-287. https://www.ncbi.nlm.nih.gov/pubmed/29853937

Response 8: The literature on coatings used for pharmaceutical applications was unknown to us. We greatly appreciate the reviewer's recommendation to consider these very interesting papers! We have added two sentences on p. 23 and included the three papers suggested by the reviewer as references 52-54.

Comment 9: In the acknowledgements, NIOSH is the National Institute for Occupational Safety and Health (“for”, not “of”).

Response 9: We thank the reviewer for catching this mistake. We have corrected the acknowledgment on p. 25.

Comments from Reviewer #2 

Comment 10: General remark about the experimental set-up:

All air samplers run simultaneously for 30 minutes. This would suggest that a high rate air sampler such as the SpinConII with a flow rate of 450 LPM would sample 13500 L in 30 minutes, compared to a low flow sampler like the NIOSH that samples only 105 L in 30 minutes with a flow rate of 3.5 LPM. Therefore, it is not surprising that within the same amount of time a high flow sampler might collect more virus. In addition, this also suggests that a low flow sampler is in competition with the strong air flow of high flow samplers. To get a more accurate impression of the performance of each sampler, the authors could have tested the air samplers separately and let the samplers draw the same amount of air (i.e. shorter or longer running time for either high or low flow samplers). This is a limitation of the current set-up that should be noted in the discussion.

Response 10: We thank the reviewer for this interesting comment, and a different way to conceive the test. Testing each sampler individually would take out interferences from other samplers, and we could control total sampling volume. However, this approach would present additional difficulties such as an impractical number of tests and would lead to some very long tests. For instance, if we decided that 15 minutes was the minimum run time for the highest flow rate sampler (anything shorter has proven to be impractical in previous tests), then the lowest flow rate sampler would require more than 32 hours to process the same volume of air. It would be difficult to maintain a constant aerosol for that long, and we would have to deal with variations in conditions among the tests. 

To address the limitation perceived by the reviewer, we have added the following sentences to the Discussion on p. 21: "The finding that lower flow rate samplers generally collected less RNA and lower titers of infectious virus is not surprising given that they are processing less air. An interesting set of experiments would be to run the samplers so that they process equal volumes of air and then determine the titers from each device."

Comment 11: Collection medium Andersen Impactor

The authors should provide more information on the collection medium used in the stages of the impactor. The performance of the impactor can be influenced by the collection medium used, also with regard to the shift in size distribution as it was observed in the present study.

Response 11: We are not sure what the reviewer means by the collection medium. The reviewer may be asking if anything is being used on top of the impactor stage. We have added a sentence on p. 8 to clarify that uncoated aluminum plates were used as the collection surface in the Andersen impactor, except for the final stage for which a filter was used. It is also possible that the reviewer is thinking about a Six-Stage Viable Andersen Cascade Impactor, in which a collection medium like agar is required. It should be clear to readers from Table 1 – as well as from other context in the manuscript – that we are using the Non-Viable Andersen impactor.

---

## [Editor Report · Decision Letter 1]

21 Dec 2020

Comparison of samplers collecting airborne influenza viruses: 1. Primarily impingers and cyclones

PONE-D-20-30871R1

Dear Dr. Raynor,

We’re pleased to inform you that your manuscript has been judged scientifically suitable for publication and will be formally accepted for publication once it meets all outstanding technical requirements.

Kind regards and Merry Christmas,

Sander Herfst

Academic Editor

PLOS ONE

---

## [Editor Report · Acceptance letter]

19 Jan 2021

PONE-D-20-30871R1 

Comparison of samplers collecting airborne influenza viruses: 1. Primarily impingers and cyclones 

Dear Dr. Raynor:

I'm pleased to inform you that your manuscript has been deemed suitable for publication in PLOS ONE. Congratulations! Your manuscript is now with our production department. 

Kind regards, 

on behalf of

Dr. Sander Herfst 

Academic Editor

PLOS ONE